



# Sensitivity analysis of a Martian atmospheric column model with data from Mars Science Laboratory

Joonas Leino[1], Ari-Matti Harri[1], Mark Paton[1], Jouni Polkko[1], Maria Hieta[1], and Hannu Savijärvi[2]

[1]Finnish Meteorological Institute, Helsinki, Finland
[2]Institute for Atmospheric and Earth System Research, University of Helsinki, Helsinki, Finland

**Correspondence:** Joonas Leino (joonas.leino@fmi.fi)

**Abstract.** An extensive sensitivity analysis was performed for a horizontally homogeneous and hydrostatic 1-D column model at the Mars Science Laboratory (MSL) location. Model experiments were compared with observations from the Curiosity Rover Environmental Monitoring Station humidity (REMS-H) device. Based on our earlier column model investigations, model surface temperature and pressure, dust optical depth ($\tau$) and column precipitable water content (PWC) were the initial

parameters that we investigated by our sensitivity analysis. Our analysis suggests that the most sensitive initial parameters for the column model temperature profile are $\tau$ and the surface temperature. The initial value of PWC does not affect the temperature profile of the model, but it is the most important parameter for the humidity profile. The initial value of $\tau$ also seems to have some effect on the humidity profile of the model. Based on our analysis, variations in surface pressure initialization are negligible for the model's humidity and temperature predictions. The model simulations are generally in good agreement with

the observations. Our analysis suggest that a slightly different shape of the model's initial humidity profile could yield better results in the predicted water vapor volume mixing ratios at 1.6 m.

## 1 Introduction

The 1-D column model, developed by the University of Helsinki (UH) and the Finnish Meteorological Institute (FMI), has been used to study the atmosphere of Mars since 1990s (Savijärvi, 1991, 1995, 1999). It has turned out to be a very useful tool for

studying the Martian atmosphere and testing new numerical algorithms (e.g., Savijärvi et al., 2016; Paton et al., 2021), as the model is extremely fast and easy to modify. This study focuses on the sensitivity analysis of the model at the Curiosity location during different seasons. We use observations of Curiosity to initialize the model and to interpret the model predictions.

     The overarching goal of this article is to better understand the inherent sensitivities in the initialization of the column model. This enhances the science return of the model when used with local in-situ observations in the analyses of the atmospheric

vertical structure and regional meteorology. The results of this study can then also be used in future studies at various landing sites.

     The dynamics of the atmospheres of Mars and Earth are very similar due to almost the same rotation rates and inclinations (Kieffer et al., 1992a; Zurek et al., 1992). Due to the dynamical similarities, several numerical atmospheric models made to study the Earth's atmosphere have been adapted to Mars, e.g. Mars Limited Area Model (MLAM, Kauhanen et al., 2008)



and 2-D Mars Mesoscale Circulation Model (MMCM, Savijärvi and Siili, 1993; Siili et al., 1999). However, the Martian atmosphere has some unique features. The surface pressure is only 500–1000 Pa, atmosphere therefore reacting very quickly to changes in radiation. In addition, the airborne dust has a strong influence on atmospheric temperatures as it absorbs solar and emits thermal radiation. Since the sensible and latent heat fluxes on Mars are very small (e.g., Savijärvi et al., 2004, Fig. 7), parametrizations used for the radiation must be accurate.

The Mars Science Laboratory (MSL) Curiosity rover landed on the floor of the Gale Crater in August 2012. It includes the Rover Environmental Monitoring Station (REMS, Gómez-Elvira et al., 2012) for measuring humidity (REMS-H, Harri et al., 2014a) and pressure (REMS-P, Harri et al., 2014b). The REMS also contains wind velocity, ground temperature, air temperature and ultraviolet sensors. The REMS-H device measures relative humidity (at the sensor) and internal sensor temperature at an altitude of 1.6 m. Here, the REMS-H temperature sensor reading is used as a proxy for the atmospheric temperature in a similar

fashion as in Savijärvi et al. (2016, 2019a, b). This also enables us to effectively compare the results of our model sensitivity study with previous analyses. The REMS-H instrument, mounted on the REMS Boom 2, onboard Curiosity can be seen from Fig. 1.

The REMS-H device humidity measurements will be re-evaluated, which will modify the calibration coefficients. Thus, the humidity values will change somewhat, but they still serve in their current form in the sensitivity analysis performed here.

The column model was used for the first time at the MSL site, when Savijärvi et al. (2015) studied diurnal temperature and moisture cycles. More advanced simulations were made in Savijärvi et al. (2016) when adsorption of moisture was included in the model. The column model experiments by Savijärvi et al. (2016, 2019a) have helped to interpret the moisture depletion in the evening and night to be caused by adsorption. Savijärvi et al. (2019a) used the model to study the diurnal moisture cycle in warm (Ls 271°) and cool (Ls 90°) seasons, while Savijärvi et al. (2019b) studied the moisture and air temperatures for three

Martian years at the MSL site. These studies showed that surface properties (thermal inertia and porosity) changed about 2.5 Martian years (MY) after landing, when the Curiosity rover started climbing Mount Sharp.

The model's diurnal adsorption process was further tested and validated by using the recalibrated Phoenix TECP data of Fischer et al. (2019), as described in Savijärvi et al. (2020) and Savijärvi and Harri (2021). That adsorption scheme is applied here.

In this study, we perform an extensive sensitivity analysis of the 1-D column model and we focus on parameters whose sensitivity has not been studied before. The structure of the model used in this study and the configuration of the analysis are described in Sect. 2. The results are presented and analyzed in Sect. 3. Finally, the results are summarized and discussed in Sect. 4.

## 2 Atmospheric column model sensitivity analysis

### 2.1 Structure of the column model

The 1-D column model, used here at the MSL site, is horizontally homogeneous and hydrostatic, therefore it does not include advections. Numerical calculations are performed in a column, which includes 29 grid points from the surface up to 50 km





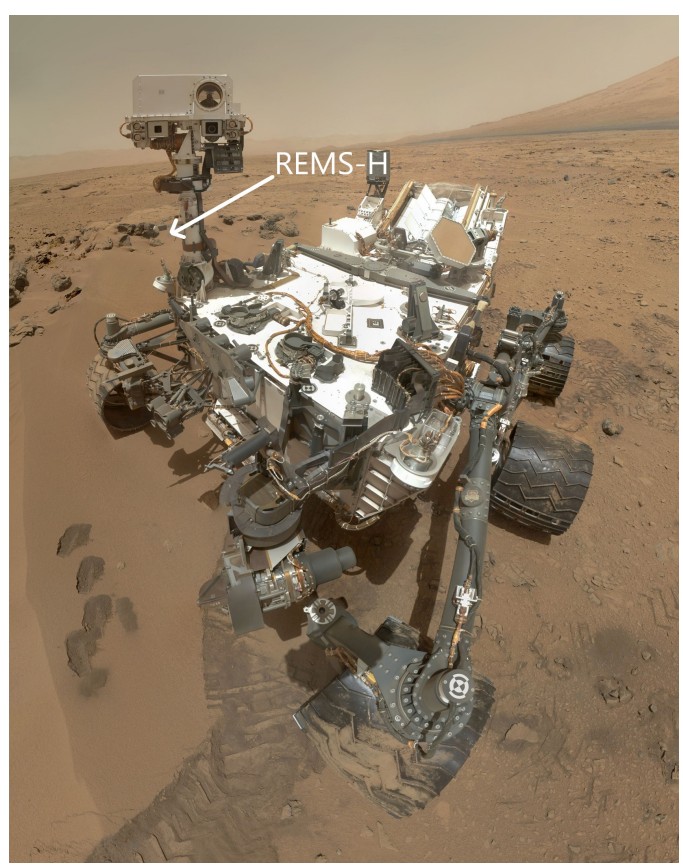

**Figure 1.** A self-portrait of the Curiosity rover produced by the Mars Hand Lens Imager, which also shows the location of the REMS-H device. (Credit: NASA/JPL-Caltech/Malin Space Science Systems)

(the lowest grid points being at 0.3, 0.7, 1.6, 3.7, 8.5 and 20 m above the surface). The predicted quantities are horizontal wind components, potential temperature and mass mixing ratios of water vapor and ice. The model and its mathematical formulations are described in Savijärvi (1999) and the radiation scheme was tested and modified in Savijärvi et al. (2004) and Savijärvi et al. (2005). In this study, we use the latest version of the column model, so the model is briefly summarized here.

Turbulence is described using a first-order closure, following Blackadar approach (Blackadar, 1962) with the asymptotic mixing length of 300 m. Diffusion coefficients depend on the local stability functions and wind shear. These stability functions are based on the Monin-Obukhov similarity theory, depending on the local bulk Richardson number. In unstable conditions, the stability function is based on the suggestion by Delage and Girard (1992), while in other conditions, the function is based on the Earth observations at midlatitudes and over the Arctic sea-ice (Savijärvi and Määttänen, 2010). The Monin-Obukhov similarity theory is used for the surface layer and the surface transfer coefficients are defined with the same stability functions as above the lowest model layer.



$CO_2$, water vapor and dust are taken into account in the radiation scheme (Savijärvi et al., 2005). An improved delta-discrete-ordinate two-stream (iDD) method is used for dust in the short-wave scheme. The dust is assumed to be well mixed, with a single scattering albedo of 0.9 and an asymmetry parameter of 0.7. The amount of airborne dust is described by the visible dust optical depth ($\tau$) at a wavelength of 0.88 $\mu$m. The $CO_2$ absorption in the short-wave scheme is based on the parametrization used by Manabe and Wetherald (1967). The $CO_2$ absorption also takes into account the radiation reflected from the surface. Rayleigh scattering and trace gases are not taken into account as their effect is extremely small based on the Spectrum Resolving Model (SRM) results in Savijärvi et al. (2005).

The long-wave radiation scheme is described using a fast broadband emissivity approach. The grey-dust approximation (with different values of the dust $\tau_{\mathrm{vis}}/\tau_{\mathrm{IR}}$ ratio for up- and down-welling fluxes) is used for the dust (Savijärvi et al., 2004). Water vapor and ice also interact with radiation and are transported by turbulence. The amount of water vapor in the atmosphere is described by the column precipitable water content (PWC).

The diffusion equation (Savijärvi, 1995), driven by the predicted ground heat flux, is used to predict the soil temperature at eight levels. Soil moisture is modeled as in Savijärvi et al. (2016, 2019a, b, 2020); Savijärvi and Harri (2021), taking into account molecular diffusion together with adsorption at the same levels as the soil temperature. The adsorption isotherm from Jakosky et al. (1997) is currently used in the model.

## 2.2 Configuration of the analysis

The REMS instrument, onboard MSL, measures pressure (P), relative humidity (RH) and temperature (T) for 5 first minutes of each hour, at an altitude of about 1.6 m. In this study, we use the median of the first measurements of RH to remove the warming effect of the sensor heads (Harri et al., 2014a) and average of the T measurements to remove turbulence. Here we use only the last measurements of P as the stable sensor (LL type) needs long warm-up time (Harri et al., 2014b). The water vapor volume mixing ratio (VMR) values are derived from the observed P, RH and T. The VMR is obtained via $\mathrm{VMR} = \mathrm{RH} \cdot e_{sat}(\mathrm{T})/\mathrm{P}$, where $e_{sat}(\mathrm{T})$ is the saturation water vapor pressure over ice as in Savijärvi et al. (2016).

The REMS-H VMR values are most accurate at minimum VMR, which usually occurs during the night at maximum RH. Thus, Figure 2 shows the REMS-H maximum RH (black) and derived VMR (purple) at the same time of sol during Martian year (MY) 32 (MSL sols 350–1018). Figure 2 also displays the daily maximum (red) and minimum (blue) REMS-H temperatures.

The warm and cool seasons are clearly displayed in Fig. 2. The coldest period occurs at around Ls 60°–120°, while the warm perihelion period is at around Ls 220°–280°. The maximum RH values are observed during the coldest time of the year, while the minimum values are during the warmest. The VMR at maximum RH reaches a minimum around Ls 60°–90°. This suggests, together with the Fig. 3 column precipitable water content (PWC) retrievals from MSL ChemCam during MY 32 (McConnochie et al., 2018), that the atmospheric moisture content at the MSL site and the near surface temperatures reach a minimum around the southern hemisphere winter solstice.

The column model experiments are performed at the MSL location (4.6°S) during the cool ($\sim$ Ls 90°, MSL sol 543) and warm ($\sim$ Ls 271°, MSL sol 866) seasons in MY 32. The hourly REMS observations, described above, are used to initialize the column model. The model's surface temperature and pressure are initialized with the sol-averaged values, calculated from the



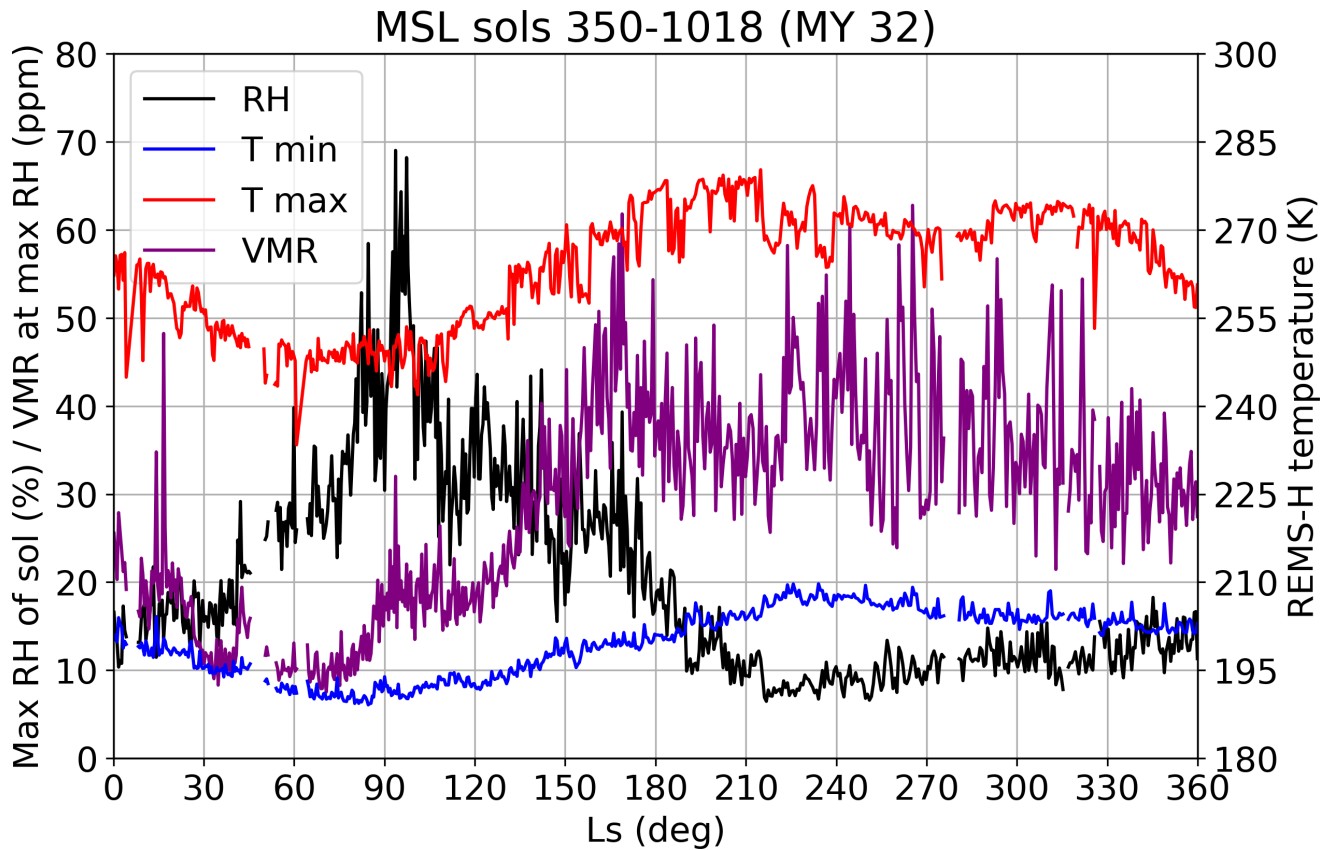

**Figure 2.** Maximum relative humidity (RH) of sol from REMS-H (black), derived volume mixing ratio (VMR) at max RH (purple) together with maximum (red) and minimum (blue) temperatures from REMS-H during Martian year (MY) 32.

hourly REMS observations of the previous sol. The temperature profile is initialized from the surface value with a lapse rate of 1 K/km and the pressure profile is calculated hydrostatically from the temperature profile.

The optical depth at 880 nm, $\tau$, is measured with the MSL Mastcam (Lemmon, 2014) and it is used to calculate the model's dust profile, which is kept constant during the simulation. The model's dust profile is well-mixed and it is given by $\tau(z) = \tau \exp(-z/H)$, where $\tau$ is the visible optical depth at the surface, $z$ is the height above the surface and $H$ is the scale height, 11 km.

MSL ChemCam passive daytime sky scans (McConnochie et al., 2018) are used to initialize the moisture profile of the
model. ChemCam measurements are used to estimate the column precipitable water content (PWC). The PWC is

$$\text{PWC} = \int_0^{p_s} q \frac{dp}{g}, \tag{1}$$





**Figure 3.** MSL ChemCam column precipitable water content (PWC) retrievals during Martian year (MY) 32.

where $p_s$ is the surface pressure, $q$ is the water vapor mass mixing ratio and $g$ is the acceleration of gravity. The model's moisture profile ($q$) is initially constant with height and it is calculated from the estimated PWC and $p_s$ using Eq. (1).

The model is initialized here with albedo of 0.18, surface roughness length of 1 cm, geostrophic wind of 10 m/s, thermal
inertia of 300 J m$^{-2}$ K$^{-1}$ s$^{-1/2}$ and porosity of 30 %. These soil properties are typical for the regolith along the Curiosity track during MY 32 (e.g., Vasavada et al., 2017). The model was then run for three sols as temperatures, winds and moistures repeat their diurnal cycles after the two-sol spin-up period.

Since the atmosphere of Mars is strongly driven by solar radiation, we choose two opposite seasons (Ls 90° and Ls 271°) in our sensitivity experiments. As radiation is extremely important in the dynamics of the thin Martian atmosphere, airborne
dust is also a key element in atmospheric models, as it absorbs solar and emits thermal radiation. Since previous studies suggest a significant effect of airborne dust in column model simulations (e.g., Savijärvi et al., 2005), the amount of dust in the atmosphere, $\tau$, is one of the parameters of our experiments. PWC is chosen because it determines the amount of water vapor in



the atmosphere and is thus a very important model initialization parameter for the diurnal water cycle, as observed by earlier column model studies (e.g., Savijärvi et al., 2016, 2019a). Apart from that, we have quite a few observations about the amount

of water vapor in the atmosphere. Therefore, it is important to study how sensitive the column model is to the initial value of PWC if we do not have direct measurements. The surface temperature is an essential variable predicted by the model and is therefore one of our parameters. The diurnal surface pressure cycle is not predicted in the model, so we choose surface pressure as the last parameter to estimate the importance of initialization accuracy.

For Ls 90°, the default initial value for $\tau$ is 0.45, 6.91 $\mu$m for PWC, 210.9 K for REMS-H mean temperature and 8.62 Pa for

REMS-P mean pressure. The corresponding parameters for Ls 271° are 0.88, 9.79 $\mu$m, 232.9 K and 9.11 Pa (cf. Figs. 2 and 3).

The reported accuracies of the REMS-P pressure and REMS-H temperature sensors are $\pm$ 3.5 Pa and $\pm$ 5 K (Martínez et al., 2017). We want to estimate the performance of the model if the initialization is not well known. As there are lots of data gaps in the measurements, some sols may miss essential observations for determining the sol-averaged T and P. The seasonal pressure

cycle is well known at the MSL site, as there are more than 3000 sols of pressure data. Thus, the sol average pressure can be estimated relatively accurately, even from some other Martian year. Since we want to see the performance of the model if the initialization is unknown, we choose the surface pressure to vary $\pm$ 10 Pa around the default value. The surface temperature is allowed to vary $\pm$ 10 K around the default value.

Dust optical depth measurements by Mastcam have an accuracy of $\pm$ 0.03 (Martínez et al., 2017), but there are only 1160

measured values during sols 33–2575. Due to the rather small number of measurements, we choose $\tau$ to vary $\pm$ 0.3 around the default value. There are even fewer PWC observations, with only 184 ChemCam PWC retrievals available during sols 230–3111. The extremely small number of measurements causes a rather large inaccuracy in the initialization of the model if there are no measurements in the vicinity of the simulated sol. The indicated precision for the ChemCam-retrieved PWC is $\pm$ 0.6 $\mu$m (McConnochie et al., 2018). The values of PWC at the MSL site are typically on the order of 10 $\mu$m (cf. 3), so we

choose the PWC to vary $\pm$ 3 $\mu$m around the default value.

## 3 Results of the sensitivity experiments

Figures 4-7 display all model experiments for the cool (Ls 90°, left panel) and the warm (Ls 271°, right panel) seasons. Modeled profiles of temperature (Figs. a and b) and humidity (Figs. e and f) are shown at 06 (black), 08 (blue), 10 (red) and 12 (orange) local time (LT) from the surface up to 5 km. The profiles show a model run with the default initial value as solid

lines, along with the simulations for the higher (+) and lower (spheres) parameter value. Modeled cycles of diurnal temperature (Figs. c and d) and VMR (Figs. g and h) at 1.6 m include model runs with default (black line), high (red line) and low (blue line) initial values together with REMS-H values (black spheres). On top of that, the VMR cycles (Figs. g and h) include the ChemCam-derived VMR (marked by x) estimated from the PWC assuming a well-mixed moisture profile (McConnochie et al., 2018).





**Figure 4.** Model results with $\tau$ being the varying parameter at Ls 90° (left) and at Ls 271° (right). Morning temperature profiles are shown in the top row ((a) and (b)), near-surface diurnal temperature cycles with hourly REMS-H observations are in the second row ((c) and (d)), morning moisture profiles are in the third row ((e) and (f)) and diurnal near-surface VMR cycles with REMS-H-derived values (spheres) and ChemCam observations (x) are in the bottom row ((g) and (h)). Profiles at 06–12 local time include high (+), default (-) and low (spheres) initial values, with each hour in a different color. Diurnal 1.6 m cycles include model simulations with high (red), default (black) and low (blue) initial values, together with MSL observations.





**Figure 5.** As Figure 4 but the initialization of the PWC is varied.

Results from the sensitivity tests are displayed in four parts, based on the varied initialization parameter. The first experiment with $\tau$ being the varying initial parameter is shown in Fig. 4. Figure 5 shows the model experiment with PWC being the



changing initial parameter. The effect of surface temperature initialization is shown in Fig. 6. Finally, the sensitivity of the model to initialization of the surface pressure is shown in Fig. 7.

In both seasons, the temperature profiles (e.g. Figs. 4a and 4b) display a strong inversion between 06 and 08 LT, while at
12 LT it is no longer present. As the atmosphere of Mars is extremely thin, the surface of Mars reacts strongly to changes in radiation. At 08 LT (blue line) convection has already started as solar radiation has started to strongly heat the surface of Mars. On top of the stronger convection in the warm season, a greater diurnal variation in temperature profiles and near-surface cycles is also easily visible.

The predicted diurnal 1.6 m T cycle is relatively close to the REMS-H-observed values in both seasons (Figs. 4c and 4d).
However, during the cool season (Ls 90°) the observed T is higher than the model's T at 14–17 LT. Also, at Ls 271°, the model's T is somewhat lower than the observations after sunrise between 09 and 11 LT. Savijärvi et al. (2016) suggested that these higher observed T are due to advection or large-scale convective cells, as these are not included in the column model.

Our simulations suggest that initialization of PWC (Fig. 5) or surface pressure (Fig. 7) does not affect model temperature profiles or 1.6 m cycles. The initialization of surface temperature affects the entire temperature profile and the shape remains
similar, as can be seen in Figs. 6a and 6b. The absolute effect appears to be slightly larger at Ls 90° compared to Ls 271°, but the difference is very small.

The amount of airborne dust (Fig. 4), however, has a big impact. At daytime (10 and 12 LT), the simulation with higher dust loading (+ markers in Figs. 4a and 4b) causes more absorption of solar radiation. Compared to the default model run (lines), this causes the atmosphere to warm above about 3 km at 12 LT (orange) and cool below that in both seasons, but the cooling
effect is slightly more pronounced at Ls 90° (Fig. 4a) than at Ls 271° (Fig. 4b), however. Since the upper atmosphere absorbs more solar radiation, the radiation does not reach the lower atmosphere as efficiently, which causes the lowest model layers to cool (Figs. 4c and 4d). At 06 LT (black), the simulation with higher dust loading (+) causes temperatures to increase in the lowest 25 m (not shown here), decrease above 25 m up to about 4.5 km, and thereafter increase again compared to the default model run (lines). A warmer atmospheric layer due to absorption by dust starts already at an altitude of 2 km at 14 LT (not
shown here). Hence, the atmosphere warms from a lower altitude due to increased solar radiation, but there is no time to heat the lowest part of the BL. After the sunset, the warmer upper atmosphere in the high-dust scenario leads the dust particles to emit more thermal radiation, which warms the lower atmosphere (Figs. 4c and 4d) and in turn cools the emitting layer.

The humidity profiles of both seasons (e.g. Figs. 4e and 4f) display a well-mixed layer in the boundary layer (BL). At 06–08 LT, the well-mixed layer is very shallow and grows thereafter due to strong convection in both seasons. As the atmospheric
moisture content in the model is higher at Ls 271°, adsorption and desorption are much stronger at Ls 271°(e.g. Fig. 4h) compared to Ls 90° (e.g. Fig. 4g, note the different scale on the y-axes). This same effect is seen by varying the initial value of PWC (Fig. 5). This initialization affects the entire atmosphere without modifying the shape of the profiles. Modified atmospheric dust loading also affects the model's humidity prediction (Figs. 4e and 4f) through radiation. Increased solar radiation near the surface in the morning drives water molecules back into the atmosphere after the nighttime adsorption. A
larger amount of available water molecules near the surface therefore increases the water content higher in the atmosphere, as





turbulence transports them vertically. The predicted near-surface VMR values start to decrease quickly in the late afternoon, when the solar radiation has weakened, e.g. Fig. 4g. This is caused by the fast decrease in temperature when adsorption begins.



**Figure 6.** As Figure 4 but the initialization of the surface temperature is varied.





**Figure 7.** As Figure 4 but the initialization of the surface pressure is varied.

The model's humidity profiles or near-surface cycles are not affected by the initialization of surface pressure (Fig. 7), but initialization of surface temperature has a small effect (Fig. 6). The water vapor mass mixing ratio and VMR values increase

with a higher initial surface temperature value, which is at least partly due to the fact that they are a function of temperature.



This temperature dependence of moisture can also affect in the model simulation with a modified dust load at a given altitude (Fig. 4e and 4f), as the initialization affects the local temperatures.

The most accurate REMS-H VMR values, derived from RH, are observed at the maximum RH. The VMR values at very low RH (< 2 %) are considered unreliable, and hence model simulations cannot be compared to these during daytime. ChemCam-

derived VMR gives here an estimate of the daytime VMR. For the model moisture quantities (RH and $q$/VMR), it is important that the predicted temperatures are accurate, as these quantities are very sensitive to temperature. The nighttime VMR derived from the REMS-H, in Figs. 4g and 4h, is relatively close to the model simulation in both seasons, but the ChemCam-derived daytime VMR is higher at Ls 90° and lower at Ls 271° than the model prediction.

If we assume that the initial PWC of the default run (from ChemCam) is correct, then the ChemCam-derived daytime VMR

(marked by x) should also be relatively accurate. In addition to this, the lowest VMR of the sol is the most accurate REMS-H observation. Thus, at Ls 90°, (Fig. 5g) the higher ChemCam-derived VMR (x) suggests that the model daytime humidity should be increased at low altitudes if the column water content is kept the same. Also, the higher REMS-H-derived VMR at about 05 LT (sphere) suggests that the nighttime VMR should be slightly higher. This is in a good agreement with the experiments made by Savijärvi et al. (2019a), as initially "low-moist layer" in the model increased 1.6 m VMR values (Fig. 4), more during

the day than at night. This is also supported by the moisture profile derived from the Mars Climate Database (MCD), Fig. 8 in Savijärvi et al. (2019a). At Ls 271° (Fig. 5h) the situation is the opposite, as during the day and early morning (about 06 LT) the moisture near the surface should be reduced. The simulation with a well-mixed initial moisture profile matched well to the observations in Savijärvi et al. (2019a), but here the modeled moisture level is somewhat higher than the observations. This is most likely due to the higher PWC (the latest data set) in our simulations compared to the lower PWC in Savijärvi

et al. (2019a). The moisture profile from the MCD (Savijärvi et al., 2019a, Fig. 8) suggests that the moisture content is more concentrated higher in the atmosphere. Such an initialization of the moisture profile could work here as well, as it should reduce the moisture content in the lower atmosphere.

## 4 Summary and discussion

The sensitivity of the 1-D column model to its initial parameters was analyzed near the equator at the MSL location in Mar-

tian year 32 during local winter and summer. Default model initialization was made using REMS-observed temperature and pressure, Mastcam-measured optical depth ($\tau$) and ChemCam-estimated column precipitable water content (PWC). We used four initial parameters in our analysis: $\tau$, PWC along with surface temperature and pressure. $\tau$ was chosen as studies of the Martian atmosphere (e.g., Savijärvi et al., 2005) indicate a major effect of dust on atmospheric temperatures through radiation. The PWC was chosen, since previous column model experiments in the Gale Crater (e.g., Savijärvi et al., 2016, 2019a) suggest

the importance of the initial PWC for the diurnal water cycle. The predicted temperature cycle is extremely important, so we also studied the effect of the initial surface temperature. The surface pressure was chosen, since the diurnal pressure cycle is not predicted in the column model.



Our simulations showed that the initialization of PWC or surface pressure does not affect the predicted diurnal temperature cycle. We found that the initial value of surface temperature affects the entire temperature profile with a slightly larger effect at Ls 90°. The amount of airborne dust had the greatest effect due to absorption of solar radiation.

The model's 1.6 m VMR cycle was close to the MSL-observed values, but they were slightly higher in the cool season and slightly lower in the warm season compared to the model prediction. An earlier study by Savijärvi et al. (2019a), large-scale model moisture profile from the MCD (Fig. 8 in Savijärvi et al. (2019a)) and our sensitivity experiments (Figs. 5g and 5h) suggest that the model's initial humidity profile at the MSL site should vary with the season to provide a better moisture prediction near the surface. This is likely due to the large-scale Hadley circulation that transports moisture in the equatorial region. It modifies the vertical distribution of moisture as well as regional atmospheric moisture content with the season (Richardson and Wilson, 2002; Navarro et al., 2014; Steele et al., 2014; Millour et al., 2017; Montmessin et al., 2017). This is an interesting result and an increased number of in-situ observations would benefit us in our research.

In addition to the shape of the initial moisture profile of the column model, the choice of adsorption/desorption scheme may play a role. This is because the adsorption/desorption is very strong on Mars, so the modeling scheme may also affect the prediction of diurnal moisture cycle directly near the surface and higher up via turbulence in the atmosphere. New missions to Mars that provide in-situ observations may help us better understand the Martian water cycle.

We found that higher moisture content during the warm season, initialization of PWC and higher near-surface diurnal variation of temperature due to lower atmospheric dust content cause higher adsorption and desorption. This also caused the water content to increase higher up in the atmosphere as a result of turbulence. We also showed that the initialization of surface pressure does not affect the predicted diurnal moisture cycle. The initialization of surface temperature, however, had a small effect, which may be due to the temperature dependence of the model's moisture quantities.

Thus, based on our sensitivity experiments, the initialization of $\tau$ and surface temperature appear to be the most important parameters for the predicted temperature profiles, while the initialization of PWC and $\tau$ looks like to be the most important parameters for the predicted humidity profiles. The varied PWC seems to be insignificant for the predicted temperatures and the modified surface pressure, in turn, looks like to be negligible for both variables. Hence, the sol-averaged surface pressure can be used even from previous years, if there are no measurements nearby. However, if the altitude of the rover is not the same, hydrostatic adjustment can be used to estimate the surface pressure. The local atmospheric dust content is, however, crucial for the model. Initialization from local observations is the most beneficial, but it can also be taken from the MCD, for example, if local observations are not available. The model's moisture profile can also be taken from the MCD if there are no local measurements to initialize the column model.

*Author contributions.* JL, A-MH, and MP planned the study; JL and A-MH performed the measurements and analyzed the data; JL wrote the manuscript draft; A-MH, MP, JP, MH and HS reviewed and edited the manuscript.



*Competing interests.* The authors declare that they have no conflict of interest.

260  *Acknowledgements.* Ari-Matti Harri, Joonas Leino and Mark Paton are thankful for the Finnish Academy grant number 310509.



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
