# Peer review of "Sensitivity analysis of a Martian atmospheric column model with data from Mars Science Laboratory"

_EGUsphere, 2023_

## Author Comment (AC1)

Dear Referee!

Thank you for your comments and improvement ideas! Please find below our responses. Referee' comments are in blue and authors' responses are in black.

The authors report experiments with some one-dimensional atmospheric column model, at the site of the Mars rover Curiosity. That includes model experiments in the warm season around solar longitude 271 and cold season around solar longitude 90. They use Curiosity data on the near-surface temperature, near-surface pressure, atmospheric dust, and atmospheric precipitable water content as initialization condition of the model. The authors study how sensitively their model results depend on these initialization conditions. This is done by increasing and decreasing the model initialization condition by certain values, re-running the model, and inter-comparing model data. Also, the authors use Curiosity data for evaluating their model experiments. The latter is done by comparisons between Curiosity data and model data.

Major revision of the manuscript is needed before publication. There are the following major comments.

Line 152-154: "On top of that, the VMR cycles (Figs. g and h) include the ChemCam-derived VMR (marked by x) estimated from the PWC assuming a well-mixed moisture profile (McConnochie et al., 2018)". Is the ChemCam-derived VMR in Figs. 4-7 (g) and (h) identical or similar to the initialization condition of the model moisture profile (Lines 109-113)? If yes, agreement between your model and ChemCam in Figs. 4-7 (g) and (h) may be trivial. And, it may not provide any new information. Please clarify (or remove the comparison with ChemCam VMR in Figs. 4-7 (g) and (h)).

Atmospheric models are always intitialized using actual observations and the model predicts future. This prediction can be compared against observations. Agreement between the model and observations provides infromation about the performance of the model. The same applies here. The column model is initialized at 00 LTST with the sol-averaged value of the previous sol. Then, the model predicts diurnal cycles for the next sol and those can be compared to observations from that sol. In addition, here we initialze the model based on only the information about the total column water content. The model then predicts the diurnal cycle of the water at each altitude. The output of the humidity profile is not well-mixed (as can be seen from Figs. 4-7 (e) and (f)). See also the answer below.

Line 150-152: "Modeled cycles of diurnal temperature (Figs. c and d) […] at 1.6 m include model runs with […] together with REMS-H values (black spheres)." Are the REMS-H temperature values in Figs. 4-7 (c) and (d) similar to the initialization condition of the model surface temperature (Line 102-103 and Line 129)? If yes, agreement between your model and REMS-H in Figs. 4-7 (c) and (d) may be trivial. And, it may not provide any new information. Please clarify (or remove the comparison with REMS-H in Figs. 4-7 (c) and (d)).

See the answer above.

To make the initialization of the column model clearer, we made small additions:

"The hourly REMS observations, described above, are used to initialize the column model **at 00 LTST.** The model's surface temperature and pressure are initialized with the sol-averaged values, calculated from the hourly **REMS-H and REMS-P** observations of the previous sol."

Line 9-10, lines 206-210 and lines 216-217: "Our analysis suggest that a slightly different shape of the model's initial humidity profile could yield better results", "… the model daytime humidity should be increased at low altitudes …", "… good agreement with the experiments made by Savijärvi et al. (2019a), as initially "low-moist layer…"", "The moisture profile from the MCD (Savijärvi et al., 2019a, Fig. 8) suggests … . Such an initialization of the moisture profile could work here as well … ." Yes, you made some interesting suggestions. Please make these suggestions happen. A revised version of the manuscript must include the following

- low-moist-layer model experiments, following Savijärvi et al. (2019a) or similar
- model initialization with the MCD moisture profile.

The authors have certainly the capacity to do that, as Dr. H Savijärvi is co-author of the manuscript.

Added new model experoments (Figures 8 and 9) and the following text:

"To test these hypotheses, column model simulations with "low-moist layer" initialization at Ls 90°, and "high-moist layer" initialization at Ls 271° were performed. These initialization profiles are shown in Fig. 8 so that the "low/high moist layer" PWC is the same as the PWC for the corresponding well-mixed profile. This "low-moist layer" assumption is based on GCM aphelion season results (e.g. Montmessin et al., 2017, Fig. 11.18), which suggests that the moisture is concentrated nearer the surface at the equatorial latitudes. However, GCM-based MCD suggests the moisture to be more well-mixed at low altitudes during the warm season (Ls 271°), and peaking at about 35 km. Hence, out "high-moist layer" assumption is based on the MCD moisture profile. "

"Figure 9 shows the simulated 1.6 m VMR cycles for Ls 90° (left panel) and Ls 271° (right panel) with REMS-H-derived VMR values (spheres) and ChemCam-derived VMR values (marked by x). Simulated cycles include "well-mixed" assumptions (red) and "low/high moist layer" assumptions. Figure 9 indeed shows that these tuned assumptions perform better compared to the "well-mixed" assumption. At Ls 90°, the "low-moist layer" initialization now matches with the REMS-H derived VMR at about 05 LTST, as well as with the ChemCam-derived VMR. Similar matches at about 06 LTST REMS-H VMR and daytime ChemCam VMR for Ls 271° is visible when using "high-moist layer" initialization."

Changed the last sentence of the abstract:

"Our additional model experiments with different shape of the model's initial humidity profile yielded better results compared to the well-mixed assumption in the predicted water vapor volume mixing ratios at 1.6 m."

Added some text to summary and discussion based on new model experiments:

"**Column model simulations with initial moisture concentrated nearer the surface ("low-moist layer") at Ls 90° and initial moisture concentrated higher in the atmosphere ("high-moist layer") at Ls 271° provided good matches with REMS-H VMR observations and ChemCam-derived VMR values.** This **seasonally varying humidity profile at the MSL site** is likely…"

"The **shape of the** model's moisture profile **should be adjusted to the location and it** can also…"

That is a major comment. More model experiments and some basic rewriting and/or extension of the article are needed.

Done.

Specific Comments

Line 2-3: "Model experiments were compared with observations from the Curiosity Rover Environmental Monitoring Station humidity (REMS-H) device." You could add the comparisons with ChemCam (unless removing them, following the major comment on line 152-154).

Added: "…(REMS-H) device **and ChemCam.**"

Line 15: "testing new numerical algorithms". Please mention briefly what numerical algorithms you mean here.

Added: "…new numerical algorithms**, such as adsorption/desorption scheme and adiabatic heating modification** (e.g., Savijärvi et al., 2016; Paton et al., 2019)…"

Line 38-39: "The REMS-H device humidity measurements will be re-evaluated, which will modify the calibration coefficients. Thus, the humidity values will change somewhat, but they still serve in their current form in the sensitivity analysis performed here." What is this information based on? Personal communication or is there any reference? Some more clarification would be helpful.

Added a sentence: "The REMS-H is designed and built at the FMI, where sensor testing and calibration are also performed (Harri et al., 2014a)."

Line 50-51: "we focus on parameters whose sensitivity has not been studied before". Please provide all such parameters.

Added a sentence: "These include surface temperature and pressure, dust optical depth (τ ) and column precipitable water content (PWC)."

Line 58-59 and line 80: "The predicted quantities are horizontal wind components, potential temperature and mass mixing ratios of water vapor and ice." (line 58-59) and "driven by the predicted ground heat flux" (line 80). Are lines 58-59 and line 80 consistent? Is the ground heat flux also a predicted quantity? Are comparisons between REMS wind data and your model possible (or not)?

The ground heat flux in the model is calculated using the predicted quantities (old lines 58-59). Therefore, it can be said that the ground heat flux is also "predicted", but it is not "predicted quantity". Comparisons between REMS wind data is not possible due to damage of the wind sensor occurred upon landing (https://atmos.nmsu.edu/data_and_services/atmospheres_data/MARS/curiosity/rems_wind.html).

Line 67-68: "and the surface transfer coefficients are defined with the same stability functions as above the lowest model layer." Please explain how stability functions depend on the height. Otherwise, that may not be immediately clear to an external reader.

Added a senrence: "These stability functions depend on the height as the bulk Richardson number depends on the buoyancy and wind shear (Louis, 1979; Stull, 1988)."

Line 74: "trace gases are not taken into account". Please specify exactly what trace gases you mean here.

Added: "…trace gases **(O2 , O3 , CO)** are…"

Line 81: "at eight levels" Are these sub-surface levels? They seem to be different from the model grid points in lines 57-58. Please clarify.

Added: "…eight **sub-surface** levels…"

Line 86: "median of the first measurements of RH". How many of the first measurements of RH? What does that mean for data accuracy?

Added: "…first **four hourly** measurements…"

Line 87: "average". Are these 5 minute averages? Do you calculate medians, arithmetic means, or something else? Please give some more details.

Added: "…and **hourly 5 minute** average…"

Line 87-88: "Here we use only the last measurements of P as the stable sensor (LL type) needs long warm-up time". How many of the last P measurements do you use? What does that mean for data accuracy? Do you calculate the median, arithmetic means, or something else? Please give some more details.

Modified the sentence: "Here we use **median of** the last **20** measurements…"

Line 102: "The model's surface temperature and pressure". Do you mean here the temperature and pressure exactly at the surface (at zero meters altitude) or at the lowest model level, which is 0.3 meters (as follows from line 58)?

We mean temperature and pressure exactly at the surface. The lowest model level is 0 meters. Added this to old line 58.

Line 101-103: "The hourly REMS observations, described above, are used to initialize the column model. The model's surface temperature and pressure are initialized with the sol-averaged values, calculated from the hourly REMS observations of the previous sol." What REMS measurements are used? Is the REMS ground temperature sensor, air temperature sensor, or REMS-H sensor temperature used for initializing the surface temperature and REMS-P for the surface pressure? See also the below comment on Line 129.

REMS-H and REMS-P observations are used. Modified the sentence:

"The model's surface temperature and pressure are initialized with the sol-averaged values, calculated from the hourly **REMS-H and REMS-P** observations of the previous sol."

Line 103-104: "lapse rate of 1 K/km". Provide evidence why a lapse rate of 1K/km is reasonable here. Is that consistent with measured or theoretical lapse rates on Mars?

Modified the sentence: "The temperature profile **at the MSL site** is initialized from the surface value with **a typical** lapse rate of 1 K/km **(Savijärvi et al., 2019a, 2020b)** and the pressure profile is calculated hydrostatically from the temperature profile."

Line 105-108: Do you use a single value or daily mean value of the Mastcam dust optical depth? Please explain.

Added: "…and **daily mean** is used…"

Line 109-113: Do you use a single value or daily mean values of ChemCam PWC or not. Please explain.

Added: "…ChemCam measurements **(single values for both sols)** are used…"

Line 113: Is the surface pressure p$_s$ based on REMS-P? Please make that clear.

Added: "… ps is the **REMS-P** surface pressure…"

Line 129: "for REMS-H mean temperature". Does that mean the REMS-H sensor temperature is used for initializing surface temperature? If yes, please state that clearly in the manuscript. Explain also why not using ground temperature sensor and air temperature sensor data. See also the above comment on Line 101-103.

Yes, REMS-H sensor temperature is used. Modified the some text in the introduction:

"Here, the REMS-H temperature sensor reading is used as a proxy for the atmospheric temperature in a similar fashion as in Savijärvi et al. (2016, 2019a, b)**, since they are estimated to deviate from the ambient temperatures by at most 1 K (Savijärvi et al., 2015).**"

Added some text:

"In this study, we use REMS-H internal temperatures instead of REMS-T air temperatures or REMS-GTS ground temperatures due to additional uncertainties of REMS-T and REMS-GTS measurements. REMS-T sensor is located only about 0.6 m above the rover deck. Thus, the heating of the rover by solar radiation and by the Radioisotope Thermoelectric Generator (RTG) may affect the air temperature measurements (Martínez et al., 2017). REMS-GTS measures the ground temperature on a small patch of nearby ground which may be different from the larger region of ground influencing the atmosphere. In addition, the field of view of the GTS is within the area of the ground heated by thermal radiation from the RTG (Hamilton et al., 2014; Martínez et al., 2017)."

Line 132-145: These paragraphs may need some rewriting. Their structure should be more logical. Take for instance the paragraph from line 132-138.

- There is the sentence "The reported accuracies of the REMS-P pressure and REMS-H temperature sensors are ± 3.5 Pa and ± 5 K (Martínez et al., 2017)." Maybe, that can be used as a minimum value for varying the model initialization (also, there could be questions such as the following. Are data uncertainties randomly distributed? Is the data uncertainty much smaller if calculating data

averages over many REMS measurements?). Maybe, you do not want to use just the minimum value. That is why you eventually select ± 10 Pa and ± 10 K.

Please think about how to rewrite the paragraphs from lines 132-145.

Paragraphs from old lines 132-145 are now rewritten. They are now as follows:

"The reported accuracies of the REMS-P pressure and REMS-H temperature sensors are ± 3.5 Pa (Martínez et al., 2017) and ± 0.1 K (Gómez-Elvira et al., 2012). **These REMS-H temperatures are estimated to deviate from the ambient temperatures by at most 1 K (Savijärvi et al., 2015). By contrast, the reported accuray for REMS-T is ± 5 K (Martínez et al., 2017) and the accuracy of the ground temperature sensor (GTS) temperatures is affected by a number of environmental variables. (Hamilton et al., 2014).**

**In this study** we want to estimate the performance of the model if the initialization is not well known. As there are lots of data gaps in the measurements, some sols may miss essential observations for determining the sol-averaged T and P. The seasonal pressure cycle is well known at the MSL site, as there are more than 3000 sols of pressure data. Thus, the sol average pressure can be estimated relatively accurately, even from some other Martian year.

Dust optical depth measurements by Mastcam have an accuracy of ± 0.03 (Martínez et al., 2017), but there are only 1160 measured values during sols 33–2575. There are even fewer PWC observations, with only 184 ChemCam PWC retrievals available during sols 230–3111. The extremely small number of measurements causes a rather large inaccuracy in the initialization of the model if there are no measurements in the vicinity of the simulated sol. The indicated precision for the ChemCam-retrieved PWC is ± 0.6 pr-µm (McConnochie et al., 2018)**, with values typically on the order of 10 pr-µm (cf. Fig. 3) at the MSL site.**

**Since we want to see the performance of the model if the initialization is unknown, we choose the sol-averaged surface pressure to vary ± 10 Pa around the default value, whereas the sol-averaged surface temperature is allowed to vary ± 10 K around the default value. In addition, variations of ± 0.3 in τ and ± 3 pr-µm in PWC are used in this study. These values are based on the sensor uncertainties but are slightly higher as we do not want to only use the minimum values.**"

Line 149: "local time (LT)". Please use local true solar time, if not done yet. Change LT to LTST (consistent with Figs. 4-7 (c)-(d) and (g)-(h); their x-axis caption is "LTST (h)").

Done. Changed all "LT" to "LTST" in the text.

Line 148-149: "at 06 (black), 08 (blue), 10 (red) and 12 (orange) local time (LT)". Please explain why selecting these times.

Added: "These times were selected because the convection is strongest during the morning hours as the sun starts to heat the surface of Mars."

Line 149: "up to 5 km". Please explain why selecting 5 km as the upper limit.

Changed the upper limit to 1 km based on the second referee comment. Added: "The upper limit of 1 km was selected to see the effect of initialization near the surface."

Line 172-182: High dust seems to give higher near-surface temperatures at night and cooler near-surface temperatures during the day. Is that correct? If yes, that may be consistent with the effects on the near-surface-temperature, known from dust storms. Any consistency with dust storms may be pointed out in the paragraph from lines 172-182 (if any). And, the paragraph may be rewritten, accordingly.

Added a new paragraph after that:

"This is consistent with known effects of the dust storms on near-surface temperature cycles. Savijärvi et al. (2020b) clearly showed an increase in near-surface temperatures at night and a decrease during the day from MSL measurements during the MY 34 global dust storm. During the same time period, Viúdez-Moreiras et al. (2020) showed the same effect of increased amount of airborne dust at the InSight location."

Line 201-202: "The nighttime VMR derived from the REMS-H, in Figs. 4g and 4h, is relatively close to the model simulation in both seasons". There seems to be some disagreement in the first half of the night, around 18-24 LTST, in Figs. 4g and 4h. More explanation is needed.

Added some text: "However, some disagreement with modeled and REMS-H derived VMRs around 18–24 LTST are visible. This is very likely related to the low RH values, as they have not yet increased enough after the extremely low daytime values. For example after the dusk at Ls 271°, observed RH is only slightly above 5 %. In contrast, observed RH during early morning hours is about 8-11 %."

**Technical Corrections**

Line10: Change "our analysis suggest" to "our analysis suggests"?

Changed.

Line 52: Change "summarized and discussed" to "discussed and summarized"?

Changed.

Line 55: Does the model have a name? That is just to make sure. If not, it is alright.

No, it's just 1-D column model.

Line 76: "The long-wave radiation scheme is described using a fast broadband emissivity approach." Does it mean that "The long-wave radiation scheme uses a fast broadband emissivity approach"?

Yes, changed.

Line 81: Change "Savijärvi et al. (2016, 2019a, b, 2020); Savijärvi and Harri (2021)" to "Savijärvi et al. (2016, 2019a, b, 2020) and Savijärvi and Harri (2021)"?

Done.

Line 91: "The REMS-H VMR values are most accurate at minimum VMR, which usually occurs during the night at maximum RH." REMS-H measures RH, not vmr. Right? Does it mean the following? REMS-H is most accurate at maximum RH. The maximum RH occurs at night and thus may coincide with minimum vmr. That may be misunderstandable. Please rephrase. Also, you could provide some more explanation for external readers, on why maximum RH and minimum vmr occur at night.

Changed the sentence: "The REMS-H is the most accurate at maximum RH, which typically occurs at night due much lower temperatures compared to daytime."

Line 92-93: "Thus, Figure 2 shows the REMS-H maximum RH (black) and derived VMR (purple) at the same time of sol during Martian year (MY) 32". You take the daily maximum of RH. Then, you convert the daily maximum of RH into VMR. Right? If so, it is self-explaining that the daily maximum of RH and its derived vmr are at the same time of sol. But, they do not occur at the same time on any sol. Right? That may be misunderstandable. Some rewriting may be needed.

Removed: "at the same time of sol"

Line 94-95: "the warm perihelion period is at around Ls 220°–280°". The red curve in Fig. 2 seems to have some dip from LS220-280.

Added som text: "In Fig. 2, the daytime maximum near surface temperatures (red curve) appear to show a small decrease during this period, due to the increased amount of airborne dust (Martínez et al., 2017). Lower daytime temperatures due to increased amount of airborne dust are shown in Sect. 3."

Line 98-99: "reach a minimum around the southern hemisphere winter solstice." Please add the related solar longitude (Ls 90°).

Added.

Line 105: "(Lemmon, 2014)". Another very recent publication may be relevant here

M.T. Lemmon, S.D. Guzewich, J.M. Battalio, M.C. Malin, A. Vicente-Retortillo, M.-P. Zorzano, J. Martín-Torres, R. Sullivan, J.N. Maki, M.D. Smith, J.F. Bell, The Mars Science Laboratory record of optical depth measurements via solar imaging, Icarus, Volume 408, 2024.

https://doi.org/10.1016/j.icarus.2023.115821.

That is just to let you know.

Changed.

Line 127: "The diurnal surface pressure cycle is not predicted in the model". External readers could have the following questions. What does that mean exactly? Why does the model need surface pressure initialization then? Please make that clear.

Modified some text: "The diurnal surface pressure cycle is not predicted in the model. However, the initialization of the surface pressure is necessary to calculate the pressure profile, which is further used in the model calculations. Hence, we choose surface pressure as the last parameter to estimate the importance of initialization accuracy."

Line 144: Change "cf. 3" to "cf. Fig 3".

Changed.

Line 151: Change "VMR" to "water vapor VMR"?

Changed.

Caption of Fig. 4: Change "default (-)" to "default (continuous line)" or similar?

Changed.

Caption of Fig. 4: Change "VMR" to "water vapor VMR"?

Changed.

Caption of Fig. 4: Change "local time" to "local true solar time"?

Changed.

Figure 4-7: Change "MSL" to "REMS-H" in the legend of sub-figures and the caption of Fig. 4?

Changed.

Line 159: External readers may need some help for seeing the temperature inversion in Figs. 4(a) and (b) (temperature increases with altitude, close to the surface, …). Please add some more details.

Added text: "… inversion**, since temperature increases with altitude close to the surface** between…"

Line 159-160: "while at 12 LT it is no longer present". The inversion is already not present at 10 LTST. Right? If right, please rephrase.

Done.

Line 161: "At 08 LT (blue line) convection has already started as solar radiation has started to strongly heat the surface of Mars." External readers may need some help. Make clear that can be seen from the lower end of the blue curve in Figs. 4-7 (a) and (b) (temperature has changed from increasing with altitude to decreasing with altitude). A close look is needed.

Added a sentence: "This can be seen from the lower end of the blue curve in Figs. 4-7 (a) and (b), since temperature has changed from increasing with altitude to decreasing with altitude."

Line 162: Change "On top of the stronger convection in the warm season" to "In addition to the stronger convection in the warm season"?

Done.

Line 183-184: "The humidity profiles of both seasons (e.g. Figs. 4e and 4f) display a well-mixed layer in the boundary layer (BL). At 06–08 LT, the well-mixed layer is very shallow and grows thereafter due to strong convection in both seasons." At 10 LTST (red curve), there seems to be a shallow well-mixed layer from ca. 100-500 meters in Fig. 4e) and 100-800 meters in Fig. 4f). That can be seen from the water vapor mass mixing ratio not changing with altitude. A similar feature is not obvious for 6 and 8 LTST (black and blue curve). Please clarify.

Added text: "At 10 LTST (red curve), there seems to be a shallow well-mixed layer from ca. 100-500 meters in Fig. 5e) and 50-750 meters in Fig. 5f). That can be seen from the water vapor volume mixing ratio (VMR) being constant with altitude. A similar feature is not obvious for 06 and 08 LTST (black and blue curve)"

Line 188-189: "Increased solar radiation near the surface in the morning". Increased solar radiation near the surface means model initialization with less dust. Right? Please say that clearly.

Changed the sentence: "...in the morning**, due to model initialization with less dust,** drives..."

Line 183-192: Please do not move back and forth between Figs. 4 and 5 in this paragraph.

Splitted the paragraph into two parts.

Line 195: "which is at least partly due to the fact that they are a function of temperature." Some more explanation is needed. Does it mean that adsorption is a function of temperature? How does it change with increasing surface temperature?

This sentence mean that water vapor mass mixing ratio and VMR are a function of temeprature. Added a sentence: "Therefore, if the temperature value increases at a given altitude, it immediately increases the mass mixing ratio and VMR values at that same altitude."

Figure 4-7: Why do you use mass mixing ratio in Figs. 4-7 e) and f) and volume mixing ratio Figs. 4-7 g) and h)?

Changed mass mixing ratio profiles in Figs. 4-7 e) and f) to VMR profiles. Added a sentence into the first paragraph of section 2.1:"In this study, model's water vapor mass mixing ratios are converted to volume mixing ratios (VMR)."

Figure 4-7: Why do the model data have some gap from 0-1 LTST in Figs. 4-7 (c)-(d) and (g)-(h)?

The model output is stored at 1,2,3… LTST

Line 205: Change "marked by x" to "marked by x in Figs. 4-7 (g) and (h)"?

Changed.

Line 208: Change "sphere" to "(sphere, Fig. 5g)"?

Changed.

Line 209: "as initially "low-moist layer" in the model increased 1.6 m VMR values". Make clear that humidity values were increased at low altitude relative to the well-mixed model experiment in Savijärvi et al. (2019a). That may not be immediately clear to an external reader.

Added: "… "low-moist layer" (**where humidity values were increased at low altitude relative to the well-mixed model experiment)** in the model..."

---

## Author Comment (AC2)

Dear Referee!

Thank you for your comments and improvement ideas! Please find below our responses.

Referee' comments are in blue and authors' responses are in black.

This paper reports a number of sensitivity analyses conducted using a one-dimensional atmospheric column model and comparisons with data acquired by REMS: pressure, near-surface temperature, and VMR at 1.6 m. The authors investigated the impact of dust optical depth, precipitable water content (PWC), surface temperature, and surface pressure values on the model results by varying these parameters in different ranges chosen based on observations. The comparison is conducted for two Ls values: 90° and 271°. Although the manuscript is generally well-written and provides new and valuable information for column model simulations on Mars, further analyses and comparison (listed below) are needed before publication.

Major comments:

-Vertical profiles of Figures 4-7: In the discussion, the authors refer to altitudes that are not shown (e.g., lines 175-182) and that are key to understanding the performance of the model. Additionally, as the comparison is with near-surface data, I would like to see the model results near the surface. If the authors want to keep the sensitivity analysis up to an altitude of 5km, then additional figures focused on the 0-1000 m range should be added as it is hard to distinguish the different curves in that range.

We decied to change the altitude from 5 km to 1 km as suggested.

-The manuscripts states that one of the most sensitive initial parameters for the column model temperature profile are the dust opacity and surface temperature. Here, I would like to see a comparison with the MCD. Also, what would be the effect if part of the aerosol opacity is due to water ice? For Ls=90° simulations, a notable % of the total opacity should be ice whose single scattering albedo is close to 1. Would it be possible to add in the model a diurnal cycle of the aerosol opacity?

The 1-D column model can only take into account local conditions. Condensation to fog and boundary layer clouds are allowed but they did not occur in any of the present integrations, due to the fairly dry equatorial Gale environment. At the moment, it is not possible to add a diurnal cycle of the aerosol opacity to the column model.

-Conclusions section: 'An earlier study by Savijärvi et al. (2019a), large-scale model moisture profile from the MCD (Fig. 8 in Savijärvi et al. (2019a)) and our sensitivity experiments (Figs. 5g and 5h) suggest that the model's initial humidity profile at the MSL site should vary with the season to provide a better moisture prediction near the surface.'. I think the authors should address this in this study. Why not taking the MCD profiles and see if the simulations improve with those model profiles? I don't think that "…the model's initial humidity profile at the MSL site should vary with the season to provide a better moisture prediction near the surface.." is demonstrated in this work, and it is not clear what this study contributes beyond the cited work. This point is also mentioned at

the end of the abstract but again no demonstration of how the humidity profile can affect is given. Please use MCD profiles and see of results change.

Added new model experements (Figures 8 and 9) and the following text:

"To test these hypotheses, column model simulations with "low-moist layer" initialization at Ls 90°, and "high-moist layer" initialization at Ls 271° were performed. These initialization profiles are shown in Fig. 8 so that the "low/high moist layer" PWC is the same as the PWC for the corresponding well-mixed profile. This "low-moist layer" assumption is based on GCM aphelion season results (e.g. Montmessin et al., 2017, Fig. 11.18), which suggests that the moisture is concentrated nearer the surface at the equatorial latitudes. However, GCM-based MCD suggests the moisture to be more well-mixed at low altitudes during the warm season (Ls 271°), and peaking at about 35 km. Hence, out "high-moist layer" assumption is based on the MCD moisture profile. "

"Figure 9 shows the simulated 1.6 m VMR cycles for Ls 90° (left panel) and Ls 271° (right panel) with REMS-H-derived VMR values (spheres) and ChemCam-derived VMR values (marked by x). Simulated cycles include "well-mixed" assumptions (red) and "low/high moist layer" assumptions. Figure 9 indeed shows that these tuned assumptions perform better compared to the "well-mixed" assumption. At Ls 90°, the "low-moist layer" initialization now matches with the REMS-H derived VMR at about 05 LTST, as well as with the ChemCam-derived VMR. Similar matches at about 06 LTST REMS-H VMR and daytime ChemCam VMR for Ls 271° is visible when using "high-moist layer" initialization."

Changed the last sentence of the abstract:

"Our additional model experiments with different shape of the model's initial humidity profile yielded better results compared to the well-mixed assumption in the predicted water vapor volume mixing ratios at 1.6 m."

-Conclusions section: The authors basically summarize the findings in the sensitivity analysis (some previously reported in previous works) but they discuss vaguely the reasons behind and do not detail the impact of their results. For instance, page 14: "We found that the initial value of surface temperature affects the entire temperature profile with a slightly larger effect at Ls 90.". Why is that? Also, is this the case at all altitudes? It is complicated to say below 1000 m from the figures.

Changed the upper limit to 1 km and added some text: "This seems to be the case at all altitudes and it is propably related to the smaller variations in the diurnal temperature cycles during the cold season compared to the warm season."

-I believe the manuscript would benefit from the addition of more data from other Ls in the comparison.

We chose 2 opposite seasons to study the sensitivity of the model as comprehensively as possible. Ls 90 is dry and cold while Ls 271 is wet and warm. Therefore, we think that these two seasons are enough to study the model's sensitivity.

-Why the authors are not included in the comparison data from MEDA??

MEDA is not used since this would require additional acceptance from the MEDA team and the main purpose of this study is to study the sensitivity of the model. With the help of this study, we can then use the model at the Perseverance landing site in the future studies.

-Include the errors in the observations, as otherwise, it is hard to figure out how well the model reproduces the data.

Uncertainties for these data are not currently available (see https://atmos.nmsu.edu/data_and_services/atmospheres_data/MARS/curiosity/rems_humidity.html). We decided to mark VMR values with very low RH (< 5 %) as gray spheres in Figs. 4–7 (g) and (h).

-It is confusing to use in the paper terms like 'profile initialized…' for parameters that do not change during the run. For the model parameters that do not change during run, please just use 'fixed profile…' or 'fix values of …'

Done.

-Section 2.2: please add information about the sampling when describing the REMS data.

Changed sentence: "The REMS instrument, onboard MSL, measures pressure (P), relative humidity (RH) and temperature (T) **at the rate of one sample per second** for 5 first minutes of each hour, at an altitude of about 1.6 m."

---

## Author Comment (AC3)

Dear Franck Montmessin!

Thank you for your comments and improvement ideas! Please find below our responses.
Your comments are in blue and authors' responses are in black.

This article presents a sensitivity study conducted with a 1D-model developed to simulate a column of atmosphere on Mars. The model includes a variety of physical processes intended to represent those to which the column is submitted at various timescales (convection, radiation, exchanges with the regolith).

The main ambition is to address the impact that some parameters have in the predictions of the model; namely temperature, relative humidity and water vapor mixing ratio. As this model has already been applied to interpret data produced by atmospheric sensors on board the Mars Science Laboratory rover (Curiosity), this work should enable the model to be used more effectively in the future, and its main limitations to be better understood.

While this paper represents a solid and valuable effort to explore the behavior of a model used to interpret Curiosity's atmospheric and surface data, it does not answer any particular scientific question and will primarily serve as a reference for the future use of the 1D model.

For this reason, the scientific contribution of the manuscript seems rather weak, while its technical value is beyond doubt.

Parameters whose impact in the predictions were studied were surface temperature and pressure, atmospheric dust and water content. The scientific contribution of this manuscript is to serve as a useful tool for studying the Martian atmosphere as well as surface-atmosphere interactions. This manuscript also describes how these parameters affect the model predictions near the surface and higher up in the atmosphere which is very important for the future studies with the model at other landing sites.

This consideration aside, the article is concise and well-structured, but suffers from several flaws which are listed below:

-It is not clear how the conclusions drawn from this study will impact on the future use of the 1D model. Some conclusions could have been avoided, as they merely confirm things already known and presented in the introduction (the negligible role of sensible and latent heat on surface temperature), while others could have been used to extend the study to arguably more representative cases (MCD). The question that should be addressed is how the findings will change the strategy for the interpretation of MSL data.

This study shows that modifications in the initial surface pressure do not affect the predictions of the model. Therefore, the initial surface pressure can be taken from the other Martian years from the same location or even from the large scale model, for example from the MCD as far as the altitude of the locations is corrected based on hydrostatic adjustment. This study also shows that it is important to initialize the atmospheric water content accurately for the humidity predictions of the model in the future. These results show that the well mixed assumption for the water content may not be the best choice to accurately predict the near surface water content. In addition, the shape of the profile can vary with the season. Therefore, the initial water content for the model should be carefully chosen on the future studies with the model on different locations as well. New model

experiments with different initial moisture profile ("low-moist layer" and "high-moist layer") were performed and they are in the revised version of this manuscript.

Revised manuscript includes 2 additional figures (Figs. 8 and 9) and some text related to new model experiments:

"To test these hypotheses, column model simulations with "low-moist layer" initialization at Ls 90°, and "high-moist layer" initialization at Ls 271° were performed. These initialization profiles are shown in Fig. 8 so that the "low/high moist layer" PWC is the same as the PWC for the corresponding well-mixed profile. This "low-moist layer" assumption is based on GCM aphelion season results (e.g. Montmessin et al., 2017, Fig. 11.18), which suggests that the moisture is concentrated nearer the surface at the equatorial latitudes. However, GCM-based MCD suggests the moisture to be more well-mixed at low altitudes during the warm season (Ls 271°), and peaking at about 35 km. Hence, out "high-moist layer" assumption is based on the MCD moisture profile. "

"Figure 9 shows the simulated 1.6 m VMR cycles for Ls 90° (left panel) and Ls 271° (right panel) with REMS-H-derived VMR values (spheres) and ChemCam-derived VMR values (marked by x). Simulated cycles include "well-mixed" assumptions (red) and "low/high moist layer" assumptions. Figure 9 indeed shows that these tuned assumptions perform better compared to the "well-mixed" assumption. At Ls 90°, the "low-moist layer" initialization now matches with the REMS-H derived VMR at about 05 LTST, as well as with the ChemCam-derived VMR. Similar matches at about 06 LTST REMS-H VMR and daytime ChemCam VMR for Ls 271° is visible when using "high-moist layer" initialization."

Changed the last sentence of the abstract:

"Our additional model experiments with different shape of the model's initial humidity profile yielded better results compared to the well-mixed assumption in the predicted water vapor volume mixing ratios at 1.6 m."

Added some text to summary and discussion based on new model experiments:

"**Column model simulations with initial moisture concentrated nearer the surface ("low-moist layer") at Ls 90° and initial moisture concentrated higher in the atmosphere ("high-moist layer") at Ls 271° provided good matches with REMS-H VMR observations and ChemCam-derived VMR values.** This **seasonally varying humidity profile at the MSL site** is likely…"

"The **shape of the** model's moisture profile **should be adjusted to the location and it** can also…"

-The role of regolith has long been an open question in the Mars water cycle community, since several Martian climate models have successfully reproduced the main features of the Mars water cycle in the absence of regolith. It is understood that the 1D model used here is based on the assumption that regolith plays an active and important role in the concentration of water vapor near the surface, which should deserve some more justification, especially in the context of contradicting results from 3D climate.

Added some text into introduction:

"The main features of the Martian water cycle may be succesfully reproduced by the climate models. However, surface observations at various locations as well as several model simulations have suggested that the near-surface moisture cycle in a diurnal timescale is dominated by the adsorption/desorption and/or salt hydration (e.g. Zent, 2014; Savijärvi et al., 2015, 2016, 2018, 2019a, 2020a; Savijärvi and Harri, 2021; Fischer et al., 2019)."

-Another unquestioned phenomenon concerns condensation and the formation of fogs. This is not mentioned in the manuscript, something that should be clarified by the authors. In particular, it would be interesting whether there is a competition between adsorption and condensation in the early morning

Fogs and boundary layer clouds are allowed to occur. However, they do not occur in any of the present integrations, due to the fairly dry equatorial Gale environment.

Added some text:

"Condensation to fog and boundary layer clouds are allowed but they did not occur in any of the present integrations, due to the fairly dry equatorial Gale environment."

-half of the graphs show a comparison between various model results as a function of altitude. Yet they should only emphasize the altitude at which the measurements are made (1.6 m) and not show T and VMR profiles up to 5 km while most of the diurnal variations occur in the first hundreds of meters .

Changed the upper limit to 1 km as suggested by the second referee.

Some text was added: "The upper limit of 1 km was selected to see the effect of initialization near the surface."

Specific comments (numbers refer to line numbers in the text):

26: 1) one of its unique features, compared to Earth, is also its 95% composition.

Added: "Martian atmosphere is mainly composed of $CO_2$ (>95 %)."

28: 2) sensible heat is negligible for the surface, but not for the atmosphere (matters for the BL)

Changed the sentence: "Since the sensibel heat flux near the surface and latent heat flux troughout the atmosphere on Mars…"

87: "and average of the T" remove of

Removed "the".

129+: PWC should be expressed in precipitable microns, pr-um.

Changed.

141: the few ChemCam observations could have been expanded by many more data from orbiters

We decided to use ChemCam observations, since these selected sols used in this manuscript had ChemCam observations and are therefore suitable to use here. In addition, ChemCam observations have been used in the previous studies by Savijärvi et al. Thus, we can compare these sensitivity experiments better with earlier studies.

Condensation to fog is allowed in the model but it did not occur.

See the response above. Added a sentence:

"The upper limit of 1 km was selected to see the effect of initialization near the surface."

3) since the text emphasizes the lack of reliability of VMR for Low RH data; RH plots should be added to let the readers see when VMR should be ignored. Alternatively, the authors could mark points when unreliable.

Very low RH values (<5 %) are now marked as gray in figures. Added a sentence: "These VMR values with very low RH (< 5 %) are shown as gray spheres in Figs. 4–7 (g) and (h)."

Added a sentence into Figure 4 caption: "Unreliable REMS-H-derived VMR values are marked as gray spheres."

202: nighttime H2O VMR for Ls 271° after dusk is not well reproduced at all. Any comment?

Added some text: "However, some disagreement with modeled and REMS-H derived VMRs around 18–24 LTST are visible. This is very likely related to the low RH values, as they have not yet increased enough after the extremely low daytime values. For example after the dusk at Ls 271°, observed RH is only slightly above 5 %. In contrast, observed RH during early morning hours is about 8-11 %."

206: It should be stressed that Chemcam cannot directly measure the H2O VMR at 1.6 m, and its value is essentially an extrapolation based on strong assumptions that the MCD, for instance, could contradict. It is clear from the graphs that H2O vmr deduced from RH measurements are made in a layer marked by a strong yet very shallow gradient. Chemcam has no sensitivity to that region of the atmosphere.

Added some text:

"This ChemCam VMR value is derived from the estimated PWC assuming well-mixed moisture profile."

New model experiments with "low-moist layer" at Ls 90 and "high-moist layer" at Ls 271 are in good agreement with the early morning REMS-H VMR and ChemCam-derived daytime VMR.

216: Since it was already discussed in previous works using the same model, why is it the MCD has not been employed to initialize the moisture profile?

This is considered in the revised version of the manuscript. See also our response to referee#1.

222: "initial" implies these parameters can evolve during the run. "Fixed" parameters see more appropriate

Apart from dust optical depth and surface pressure, these parameters evolve during the model run. We decided to remove "initial".

---

## Author Response (AR2)

Dear Referees and topic editor!

Thank you for your comments and improvement ideas! Please find below our responses. Referee' comments are in blue and authors' responses are in black.

Dear Dr. Leino, dear co-authors,

I thank you for considering the comments of the referees and for making adequate modifications to the text. It is my pleasure to inform you that after the second revision both opponents require only minor additional modifications (please, see suggestions below). Once these corrections have been made, your paper will be recommended for publication. Please make these edits as soon as possible and send back the text with the modifications highlighted.
Kindest regards

D. Buresova

**Referee 1**

The manuscript has improved compared to the previous version thanks to the authors' work, but there are still a few points that need to be addressed before publication.

-In my initial review, I suggested showing the results in more detail in the 0-1 km range, as it was difficult to distinguish between the different curves in that range, and the paper discusses important results there. I did not mean to modify the upper limit of the figure from 5 km to 1 km, but rather to include an additional figure specifically focusing on the 0-1 km range. Additionally, it would be beneficial for readers to have access to the figures from the previous version of the manuscript that extend up to 5 km, perhaps as supplementary material.

Added figures that extend up to 5 km in the appendix. Modified sentences:

"The upper limit of 1 km was selected to see the effect of initialization near the surface. **Appendix A shows the profiles up to 5 km.**"

"Compared to the default model run (lines), this causes the atmosphere to warm above about 3 km at 12 LTST (**shown in Fig. A1**)..."

"...decrease above 25 m up to about 4.5 km **(Fig. A1)**, and..."

"That can be seen from the water vapor volume mixing ratio (VMR) being constant with altitude **(see also Figs. A2e and A2f).**"

-Regarding the errors in the observations, I pointed out the need to include uncertainties in the data to assess how well the model reproduces it. However, the authors mentioned that uncertainties for these data are currently unavailable. If this is the case, sentences like 'The model's 1.6 m VMR cycle was close to the MSL-observed values, but they were slightly higher in the cool season and slightly lower in the warm season compared to the model prediction' need to be justified. If uncertainties cannot be included when comparing data vs. model, it should be acknowledged that this comparison is limited by the absence of uncertainties.

Modified the text a little:
"The predicted diurnal 1.6 m T cycle is relatively close to the REMS-H-observed values in both seasons (Figs. 4c and 4d)**, but this comparison is limited by the absence of uncertainties.**"

"The model's 1.6 m VMR cycle was close to the MSL-observed values, but they were slightly higher in the cool season and slightly lower in the warm season compared to the model prediction. **However, this comparison is slightly limited by the absence of uncertainties.**"

-Regarding the use of MEDA data, it's important to note that acceptance from the MEDA team is not necessary for using data that has already been published or is available in the PDS. I encourage the authors to consider extending this work using MEDA data for future studies.

We will consider this for future studies.

-The authors mentioned that 'at the moment, it is not possible to add a diurnal cycle of the aerosol opacity to the column model.' In this case, it would be beneficial to include a discussion in the manuscript about how such an aerosol opacity cycle would affect the simulations, based on the aerosol opacity cycles observed in Gale crater.

Added some text in the Summary and discussion:

"A diurnal cycle of aerosol opacity has been observed in the Gale Crater (e.g., Lemmon et al., 2024). Nonetheless, this cycle is not simulated in the column model, but we assume that it should affect in a similar fashion as in the sensitivity experiment with varying tau. Higher opacity during the day would decrease near-surface temperatures as the atmosphere absorbs more solar radition. By contrast, during the nighttime higher amount of aerosols in the atmosphere would increase near-surface temperatures due to increasing thermal radiation. In the future column model simulations, it would be interesting to test this feature in practice."

**Referee 2**
The authors have significantly improved the manuscript.
The reviewer has only minor comments.

Line 29: There may be some typo. It should be "sensible heat flux". Right?
Done.

Line 53-55: "The main features of the Martian water cycle may be succesfully reproduced by the climate models. However, surface observations at various locations as well as several model simulations have suggested that the near-surface moisture cycle in a diurnal timescale is dominated by the adsorption/desorption and/or salt hydration". Why do you use the word "however" here? That is not fully clear.

Removed.

Line 103: " due much lower temperatures". Do you mean "due to"?

Done.

Line 120-121 and line 158-159: "measurements. (Martínez et al., 2017)" and "variables. (Hamilton et al., 2014)". Shall the citation be before the full stop?

Yes, done.

Line 204-215: Please check whether some rearranging of the information in this paragraph is needed. Possibly, you jump forth and back between different LTSTs. Just as a thought.

Decided to split the parapgraph into two paragraphs to make it more clear to the readers.

Line 234: "The model's humidity profiles or near-surface cycles are not affected by the initialization of surface pressure (Fig. 7)". Isn't there a small effect in Figures 7e) and f)?

Actually, yes there is a very small effect. Modified the text:

"The model's humidity profiles and near-surface cycles **are affected a little** by the initialization of surface pressure (Fig. 7) and temperature (Fig. 6). **The very small effect by the initialization of surface pressure to water vapor VMR is very likely caused by the fact that VMR value depends on the pressure value (VMR = RH · esat (T)/P). Moreover,** the water vapor…"

Made also a small modification to the abstract:
"Based on our analysis, variations in surface pressure initialization are negligible for the model's temperature **and almost negligible for the model's** humidity predictions."

Modification to Summary and discussion:
"We also showed that the initialization of **surface pressure and temperature have a very small effect on** the predicted diurnal moisture cycle. **This is very likely due to the temperature and pressure dependence of the model's moisture quantities.**"

"...looks like to be **almost** negligible for both variables..."

Line 236-238: "which is at least partly due to the fact that they are a function of temperature. Therefore, if the temperature value increases at a given altitude, it immediately increases the mass mixing ratio and VMR values at that same altitude." Isn't VMR temperature-independent and pressure-independent? For instance, think of the gas in a balloon. The balloon expands as it rises. But, the mixing ratio of different gases in the balloon remains constant. Please check.

See the equation in the first paragraph of Section 2.2 for VMR. Here are also some formulas from Savijärvi et al. (2016):
$$e = RH \times e_{sat},$$
where $e$ is the water vapor partial pressure. The water vapor mass mixing ratio q is given as follows:
$$q = \frac{\varepsilon e}{p},$$
where $\varepsilon=18/44$ (ratio of the molecular weight of water to that of dry air) and $p$ is the pressure. In addition, VMR can be calculated from q:

VMR=q/ε

Therefore, e depends on temperature as $e_{sat}$ depends on temperature (Savijärvi et al. 2016, eq. 1). Thus, also q and VMR depend on temperature.

Line 257: Please add that this is their Figure 4, not yours.

Added.

Line 270: "out". Do you mean "our"?

Yes, done.